# Stabilize to Act: Learning to Coordinate for Bimanual Manipulation

**Jennifer Grannen, Yilin Wu, Brandon Vu, Dorsa Sadigh**
Stanford University, Stanford, CA
`jgrannen@stanford.edu`

**Abstract:** Key to rich, dexterous manipulation in the real world is the ability to coordinate control across two hands. However, while the promise afforded by bimanual robotic systems is immense, constructing control policies for dual arm autonomous systems brings inherent difficulties. One such difficulty is the high-dimensionality of the bimanual action space, which adds complexity to both model-based and data-driven methods. We counteract this challenge by drawing inspiration from humans to propose a novel role assignment framework: a *stabilizing* arm holds an object in place to simplify the environment while an *acting* arm executes the task. We instantiate this framework with BimanUal Dexterity from Stabilization (BUDS), which uses a learned *restabilizing classifier* to alternate between updating a learned stabilization position to keep the environment unchanged, and accomplishing the task with an acting policy learned from demonstrations. We evaluate BUDS on four bimanual tasks of varying complexities on real-world robots, such as zipping jackets and cutting vegetables. Given only 20 demonstrations, BUDS achieves 76.9% task success across our task suite, and generalizes to out-of-distribution objects within a class with a 52.7% success rate. BUDS is 56.0% more successful than a unstructured baseline that instead learns a BC stabilizing policy due to the precision required of these complex tasks. Supplementary material and videos can be found at https://tinyurl.com/stabilizetoact.

**Keywords:** Bimanual Manipulation, Learning from Demonstrations, Deformable Object Manipulation

## 1 Introduction

Bimanual coordination is pervasive, spanning household activities such as cutting food, surgical skills such as suturing a wound, or industrial tasks such as connecting two cables. In robotics, the addition of a second arm opens the door to a higher level of task complexity, but comes with a number of control challenges. With a second arm, we have to reason about how to produce coordinated behavior in a higher dimensional action space, resulting in more computationally challenging learning, planning, and optimization problems. The addition of a second arm also complicates data collection—it requires teleoperating a robot with more degrees of freedom—which hinders our ability to rely on methods that require expert bimanual demonstrations. To combat these challenges, we can draw inspiration from how humans tackle bimanual tasks—specifically alternating between using one arm to *stabilize* parts of the environment, then using the other arm to *act* conditioned on the stabilized state of the world.

Alternating stabilizing and acting offers a significant gain over both model-based and data-driven prior approaches for bimanual manipulation. Previous model-based techniques have proposed planning algorithms for bimanual tasks such as collaborative transport or scoping [1, 2, 3], but require hand-designed specialized primitives or follow predefined trajectories limiting their abilities to learn new skills or adapt. On another extreme, we turn to reinforcement learning (RL) techniques that do not need costly primitives. However, RL methods are notoriously data hungry and a high-dimensional bimanual action space further exacerbates this problem. While simulation-to-real transfer techniques offer an appealing alternative [4, 5, 6, 7], a key component of bimanual tasks is

closed-chain contacts and high-force interactions (consider cutting or connecting cables), which are hard to simulate and widen the gap with reality [8, 9, 10]. A more promising data-driven approach is learning from demonstration. However, collecting high-dimensional bimanual demonstrations is difficult as simultaneously controlling two high-degree freedom arms often requires specialized hardware or multiple operators [11, 12, 8, 13, 14]. The increased dimensionality of the action space also necessitates significantly more data, especially for more precise or dexterous tasks [15].

Our insight about how humans *iterate between stabilizing and acting* presents a way to overcome these challenges. In tasks such as plugging in a phone or cutting a steak, a stabilizing arm holds an object (e.g. the phone or steak) stationary to simplify the environment, making it easier for the acting arm to complete the task with high precision. Factoring control across stabilizing and acting additionally offers benefits for data collection; the role-specific policy can be learned independently for each arm, bypassing the need for bimanual demonstrations. Adjusting a stabilizing position iteratively as the acting arm progresses enables even more expressive and generalizable behavior. For example, a fork should skewer a steak at different points depending on where the knife cuts.

*Thus, the key insight driving this work is that to enable sample-efficient, generalizable bimanual manipulation, we need two roles: a **stabilizing** arm stabilizes an object to simplify the environment for an **acting** arm to perform the task.*

We propose BimanUal Dexterity from Stabilization (BUDS), a method that realizes this coordination paradigm by decomposing the bimanual problem into two single-arm problems: learning to stabilize and learning to act. The stabilizing policy decides *where* to stabilize in the scene and *when* to adjust, while the acting arm learns to perform the task in this simpler environment. For example when cutting a steak, our *stabilizing policy* learns where to hold a steak and when to adjust so the steak remains stationary while the *acting policy* makes the cut.

To learn where to stabilize, we use a vision-based system that takes an environment image as input

Figure 1: **BUDS: BimanUal Dexterity from Stabilization**: BUDS is a bimanual manipulation framework that uses a novel stabilizing and acting role assignment to efficiently learn to coordinate. For the stabilizing role, BUDS learns a Stabilizing Position Model (1) to predict a point to hold stationary using a noncompliant controller (2). In this simplified environment, BUDS learns to act from single-arm demonstrations (3). Combined, these two actions comprise a bimanual policy (4). Finally, at every timestep, BUDS's Restabilizing Classifier (5) predicts whether the stabilizing position is still effective or needs to be updated.

and outputs a stabilization keypoint position. We then learn a restabilizing classifier that determines from images when the stabilizing keypoint is no longer effective and needs to be updated. We deploy this stabilizing policy while collecting *single-arm* trajectory demonstrations for an acting policy to sidestep the need for a precise and expensive bimanual demonstration collection interface. Using these demonstrations, the acting arm learns a policy via imitation learning to accomplish the task in this simplified, stationary environment. We demonstrate the efficacy of this paradigm on four diverse, dexterous manipulation tasks on a physical UR16e dual-arm platform. BUDS achieves 76.9% success, and outperforms an unstructured baseline fully learned from expert trajectory demonstrations by 56.0%. Additionally, BUDS achieves 52.7% when generalizing to unseen objects of similar morphology (e.g. transferring a cutting policy trained on jalapeños to cutting zucchini and celery).

Our contributions are: (1) A paradigm for learning bimanual policies with role assignments, where the stabilizing arm stabilizes the environment and reduces the non-stationarity present while an acting arm learns to perform a task allowing in a simpler learning setting, (2) A framework for

collecting bimanual demonstrations that bypasses the need for a dual-arm interface by collecting demonstrations for the stabilizing and acting roles independently, and (3) A system, BUDS, that instantiates this paradigm to learn a centralized bimanual policy to perform a broad range of tasks.

## 2 Related Work

In this section, we describe the current data-driven and model-based methods available for bimanual manipulation tasks, along with prior work using ideas of stabilizing for manipulation.

**Learning-based Bimanual Manipulation.** A recurring challenge throughout bimanual manipulation is the high-dimensionality of the action space. This appears both in reinforcement learning (RL) and imitation learning (IL) works [16, 11, 17, 18, 19, 20, 21]. Prior multi-agent coordination works have considered shrinking the high-dimensionality of the problem by using a second agent stabilizing a latent intent [22], however learning a stabilizing policy and a latent intent mapping both require a significant amount of data that is not realistic for physical robot manipulation tasks.

RL methods for learning high-frequency bimanual control policies can require a large number of samples and many hours of robot time, which makes simulation to real policy transfer an appealing approach [21, 7, 6]. However, sim-to-real approaches are limited to settings where the sim-to-real gap is small, which precludes many contact-rich bimanual tasks such as zipping a zipper or cutting food [8, 9, 10]. Instead, works in both RL and IL settings have proposed using parameterized movement primitives to reduce the action space, and have achieved reasonable success on tasks such as opening a bottle and lifting a ball [17, 19, 20, 23, 24, 25, 26, 27, 28, 29, 15, 30]. However, these movement primitives greatly limit the tasks achievable by the method as they often require costly demonstrations or labor-intensive hard-coded motions for each task-specific primitive. Additionally, learning from demonstrations in bimanual settings is difficult as teleoperating two high-degree-freedom robots or collecting kinesthetic demonstrations on both arms simultaneously is challenging and sometimes impossible for a single human and may require specialized hardware [16, 11, 31, 12, 8, 13]. Recent works have demonstrated more effective interfaces for data collection in a bimanual setting, but these interfaces are limited to specific hardware instantiations and would still require large amounts of expert data to learn a high dimensional policy [14]. To avoid the need for expert bimanual demonstrations, we use a novel stabilizing paradigm to decouple the arms' policies and learn a role-specific policy for each arm from single-arm demonstrations. This added structure also brings down the dimensionality of the large action space in a task-agnostic way.

**Model-based Bimanual Manipulation.** The majority of model-based bimanual manipulation methods are limited to using planning and constraint solving methods to jointly move or hold a large object [32, 33, 34, 35, 12, 2, 1, 36]. Bersch et al. [37] and Grannen et al. [3] present systems using a sequence of hard coded actions for folding a shirt and scooping food respectively. However, as tasks become more complex, the primitives required also become more unintuitive and costly to hand-design. We instead learn a control policy from single-arm demonstrations, avoiding the need for labor-intensive hand-coded primitives while performing dexterous bimanual manipulation tasks.

**Stabilizing for Manipulation.** Stabilizing and fixturing can yield large benefits in a manipulation context by providing additional steadiness for high precision tasks and unwieldy object interactions. Early works in industrial robotics have proposed planners for autonomous fixture placement that reason about friction forces [38] or use CAD model designs [39] to add structure to the environment. More recent works have used additional fixture arms or vises to bootstrap sample efficiency [40] or avoid robot force and torque limits [41]. Similarly, Chen et al. [42] consider a collaborative setting—an assistive robot arm reasons about forces to hold a board steady for a human to cut or drill into. The addition of an assistive stabilizing role naturally points towards a bimanual setting, and indeed many bimanual manipulation works implicitly use a stabilizing role in their designs [23, 11, 3, 21]. Holding food in place while cutting is, perhaps, an obvious application of stabilizing, and this assistance is critical for overcoming the highly variable geometries and dynamics of food [43, 44, 45]. While prior stabilizing works are limited as a task-specific systems, we propose a *general* bimanual system that learns from demonstrations how to stabilize and act for a variety of tasks.

# 3 Stabilizing for Bimanual Tasks

Given a set of expert demonstrations, we aim to produce a bimanual policy for executing a variety of manipulation tasks, such as capping a marker or zipping up a jacket. We formulate each bimanual task as a sequential decision making problem defined by components $(\mathcal{O}, \mathcal{A})$. Each observation $o_t$ comprises an RGB image frame $f_t \in \mathbb{R}^{H \times W \times 3}$ and the proprioceptive state of each arm $p_t \in \mathbb{R}^{14}$. $\mathcal{A}$ is the action space for the two robot arms containing 14 degrees of freedom (DoF) joint actions $a_t$. We define $a_t = (a_t^s, a_t^a)$, where $a_t^s, a_t^a \in \mathbb{R}^7$ are the stabilizing and acting arm actions respectively. We are in a model-free setting, and make no assumptions on the unknown transition dynamics.

To perform these bimanual tasks, we use a bimanual manipulator operating in a workspace that is reachable by both arms, along with a standard $(x, y, z)$ coordinate frame in this workspace. We use depth cameras with known intrinsics and extrinsics, which allows us to obtain a mapping $(f_x, f_y)$ in pixel space to a coordinate $(x, y, z)$ in the workspace, which we later refer to as a keypoint.

To learn our bimanual policies, we first assume access to a set of expert bimanual demonstrations $\mathcal{D}$, and later relax this assumption to two sets of expert *unimanual* demonstrations $\mathcal{D}^a$ and $\mathcal{D}^s$ to avoid the challenges of collecting bimanual demonstrations. Each demonstration is a sequence of observation, action pairs that constitute an expert trajectory. First, we consider bimanual demonstrations $[(o_1, a_1^s, a_1^a), (o_2, a_2^s, a_2^a), \dots] \in \mathcal{D}$ to discuss the challenges of learning a Monolithic policy. Next, we pivot to decoupling the bimanual policy with two unimanual datasets: $[(o_1, a_1^a), (o_2, a_2^a), \dots] \in \mathcal{D}^a$ and $[(o_1, a_1^s), (o_2, a_2^s), \dots] \in \mathcal{D}^s$.

## 3.1 Monolithic 14-DoF Policy

Let us first consider learning a monolithic 14-DoF policy $\pi_\theta(a_t^s, a_t^a | o_t)$ parametrized by $\theta$ via behavioral cloning, which takes an observation $o_t$ as input and outputs a bimanual action $(a_t^s, a_t^a)$. We aim to find a policy that matches the expert demonstrations in $\mathcal{D}$ by minimizing this supervised loss:

$$\mathcal{L}(\theta) = -\mathbb{E}_{(o, a^s, a^a) \sim \mathcal{D}} \log \pi_\theta(a^s, a^a | o). \tag{1}$$

While this is feasible in theory, in practice learning policies in this way is highly dependent on clean and consistent demonstration data for both arms acting in concert. However, as mentioned in Section 2, collecting such data is challenging and these difficulties are further exacerbated for precise and dexterous tasks. Motivated by stabilizing structures across many bimanual tasks, we sidestep these challenges by utilizing a task-agnostic role-assignment while learning bimanual policies.

## 3.2 Stabilizing for Reducing Control Dimensionality

We observe that a wide variety of human bimanual tasks leverage a similar paradigm: one arm *stabilizes* objects in the scene to simplify the environment while the other arm *acts* to accomplish the task. We translate this observation into a generalizable robotics insight: assign either a stabilizing or acting role to each arm to specify a coordination framework. Thus, we decompose our bimanual policy $\langle a_t^s, a_t^a \rangle \sim \pi(\cdot | o_t)$ into two role-specific policies: a stabilizing policy $a_t^s \sim \pi_{\theta^s}^s(\cdot | o_t, a_t^a)$ and an acting policy $a_t^a \sim \pi_{\theta^a}^a(\cdot | o_t, a_t^s)$. These policies are co-dependent; we aim to disentangle them.

Given these roles, we make a crucial insight: for a given acting policy subtrajectory $(a_i^a, a_{i+k}^a)$, there exists a single stabilizing action $\bar{a}^s$ that works as a "fixture" for holding constant a task-specific part of the environment. For example, consider the role of a fork pinning a steak to a plate to facilitate cutting with the knife. These stabilizing fixtures act to reduce the dimensionality of the control problem for the other arm, as the environment is less susceptible to drastic changes. We characterize this constant task-specific region with a learned task-relevant representation $\phi : \mathcal{O} \mapsto \mathbb{R}^j$ for some $j$, and we later instantiate a stabilizing fixture $\bar{a}^s$ with a keypoint representation in Section 4.1 and execute non-compliant motions at this keypoint. Finally, we isolate our stabilizing policy $\pi_{\theta^s}^s(a_t^s | \phi(o_{t-1}), \phi(o_t))$ from the acting policy with a loss that penalizes the expected change in a task-relevant region of the environment:

$$\mathcal{L}(\theta^s) = \sum_{t=0}^{k} \mathbb{E}_{a_t^a \sim \pi_{\theta^a}^a(\cdot | o_t, a_t^s)} ||\phi(o_t) - \phi(o_{t-1})||. \tag{2}$$

Given the stabilizing action $\bar{a}_t^s \sim \pi_{\theta^s}^s(\phi(o_{t-1}), \phi(o_t))$, we obtain an acting action $a_t^a \sim \pi_{\theta^a}^a(\cdot|o_t, \bar{a}_t^s)$. This stabilizing action is valid for $k$ timesteps, afterwards which the stabilizing fixture must be updated. To obtain this variable length $k$, we threshold the change in $\phi(o_{i+n})$ from the initial observation $\phi(o_i)$ to indicate when a stabilizing fixture is no longer effective:

$$k = \inf\{n : n \geq i \text{ and } ||\phi(o_{i+n}) - \phi(o_i)|| > \epsilon\} \tag{3}$$

In practice, we instantiate the task-relevant representation to be stabilized $\phi(o_t)$ as a keypoint model learned from expert demonstrations (using a learned mapping from an image to a keypoint $f^k : i \mapsto w^s$). We do not solve Eq. (2) but instead utilize a noncompliant controller to hold this point stationary over time (see Section 4.1). Given a stabilizing fixture that is effective for acting actions $a_{[i, i+k]}^a$, we additionally learn a restabilizing classifier $f^r(o_t) = \{0, 1\}$ that determines when $k$ has been surpassed and a new learned stabilizing action should be predicted. We describe this implementation further in Section 4 and show in our experiments in Section 5 that this approximation holds.

## 4  BUDS: BimanUal Dexterity from Stabilization

We describe BimanUal Dexterity from Stabilization (BUDS), which instantiates the stabilizing and acting role assignments in Section 3. As shown in Fig. 1, we learn a model for each role: $f_\theta^k$ for stabilizing and $\pi_\phi^a$ for acting, parameterized by weights $\theta$ and $\phi$. We also learn a restabilizing classifier $f_\psi^r$, parameterized by weights $\psi$. All models are learned from human-annotated images or single-arm teleoperated robot demonstrations, avoiding the difficulties of collecting bimanual demonstrations. All labels and demonstrations are consistent across both arms for any given image.

### 4.1  Learning a Stabilizing Policy

From Section 3, we aim to find a stabilizing policy $\bar{\pi}^s(s(o_{t-1}), s(o_t)) = \bar{a}_t^s$. Specifically, we aim to learn a task-specific representation $s$ to be stabilized. We observe that when humans stabilize in bimanual tasks, they *hold a point stationary over time*. Thus, $\mathcal{D}^s$ contains two action types: stationary or zero-actions that

**Algorithm 1** Stabilizing with BUDS

1: **while** Task Incomplete **do**
2:     $\hat{w}_t^s = f_\theta^k(o_t)$
3:     **while** $f_\psi^r(o_t) = 0$ **do**
4:         $a_t^s = \pi^s(\hat{w}_{t-1}^s, \hat{w}_t^s)$   $\triangleright$ $\{a_t^s : \hat{w}_t^s \simeq \hat{w}_{t-1}^s\}$
5:         $a_t^a \sim \pi_\phi^a(o_t, a_t^s)$
6:         Execute $a_t^s, a_t^a$. Observe $o_{t+1}, f_\psi^r(o_{t+1})$.
7:     **end while**
8: **end while**

hold a point in place and transient actions that move between stabilizing positions. Additionally, this observation implies $s$ can be instantiated as a mapping from an observation $o_t$ to a stabilization position $w_t^s$. We decompose the stabilizing role into two parts: (1) selecting a stabilization position $w^s$ to hold stationary and (2) sensing when to update the stabilization position (as in Eq. (3)).

We parameterize $w^s$ as a keypoint on an overhead image of the workspace. We use a ResNet-34 [46] backbone to learn a mapping $f_\theta^k : \mathbb{R}^{640 \times 480 \times 3} \rightarrow \mathbb{R}^{640 \times 480}$, which takes as input an overhead image and outputs a Gaussian heatmap centered around the predicted stabilizing keypoint $\hat{w}^s$. This mapping is learned from the stationary actions in the demonstration data $\mathcal{D}^s$, indicating that the arm is at a stabilizing position in this demonstration. In practice, we bypass the need for full trajectory demonstrations and provide supervision in the form of keypoint annotations. Given $\hat{w}^s$ and a depth value from the overhead camera, a non-compliant controller grasps this 3D point and holds it stationary. Thus, we approximate the stabilizing action $a_t^s$ with the action that keeps the keypoint stationary, i.e., $\hat{w}_t^s \approx \hat{w}_{t-1}^s$. We can then write $\pi^s(s(o_{t-1}), s(o_t))$ as $\pi^s(\hat{w}_{t-1}^s, \hat{w}_t^s)$, a function of two consecutive keypoints learned from demonstrations: $\hat{w}_t^s = f_\theta^k(o_t)$. The learned keypoint mapping $f_\theta^k$ is trained with a hand-labelled dataset of 30 image and keypoint pairs, where the keypoint is annotated as the stabilizing keypoint $w_t^s$ for the image. We fit a Gaussian heatmap centered at the annotation with a standard deviation of 8px. This dataset is augmented 10X with a series of label-preserving image transformations [47] (see Appendix A). From this dataset, $f_\theta^k$ learns to predict the keypoint $\hat{w}^s$ for the stabilizing policy to hold stationary.

To determine when to update $w^s$, we close the feedback loop by learning a restabilizing classifier $f_\psi^r : \mathbb{R}^{640 \times 480 \times 3} \rightarrow \{0, 1\}$ that maps input workspace images to a binary output indicating whether

or not to update $w^s$. This mapping is learned from the transient actions in the demonstration data $\mathcal{D}^s$—indicating that the stabilizing positions at these states need to be updated. In practice, we forgo using full trajectory demonstrations for supervision in the form of binary expert annotations. We instantiate $f_\psi^r$ with a ResNet-34 [46] backbone and train this classifier with an expert-labelled dataset of 2000 images. For each rollout, an expert assigns when in the rollout a new stabilizing position $w^s$ is needed; the preceding images are labelled 0 while the following images are labelled 1. This dataset is augmented 2X with affine image transformations (See Appendix A for details). $f_\psi^r$ learns to predict a binary classification of when the stabilizing point is no longer effective and needs to be updated with $f_\theta^k$. Together, $f_\theta^k$ and $f_\psi^r$ define a stabilizing policy $\pi^s$ as outlined in Algorithm 1.

## 4.2 Learning an Acting Policy

Given a stabilization policy $\pi^s(\hat{w}_{t-1}^s, \hat{w}_t^s)$, an acting policy $\pi_\phi^a$ learns to accomplish the task in a simpler stationary environment. We instantiate $\pi_\phi^a$ with a BC-RNN architecture that is trained on 20 single-arm demonstrations. A expert teleoperates the acting arm using a SpaceMouse [48], a 3D joystick interface During data collection, the stabilizing arm is assumed to be in the expert-labelled stabilizing position $w^s$ and the environment is in a simplified state. $\pi_\phi^a$ optimizes the standard imitation learning loss as defined in Eq. (1), and we refer the reader to Appendix A for more details.

To further increase sample efficiency, we assume that our expert acting demonstrations start from a pre-grasped initial position. To achieve this pre-grasped position, we train an optional grasping keypoint model $f^g$ for the acting policy that maps input workspace images $i_t \in \mathbb{R}^{640 \times 480 \times 3}$ to a Gaussian heatmap centered around the grasp point. This grasping model is instantiated with the same ResNet-34 [46] and dataset parameters as used for the stabilizing keypoint model $f_\theta^k$. The acting arm moves to the keypoint position in a fixed orientation, and grasps to begin the task.

## 5 Experiments

We validate BUDS on four diverse bimanual tasks. We use two UR16e arms each with a Robotiq 2F-85 gripper, mounted at a $45°$ angle off a vertical mount, 0.3m apart. We use a RTDE-based impedance controller [49] and associated IK solver operating at 10Hz on an Intel NUC. End effectors move along a linear trajectory between positions. All grasps use a grasping force of 225N and a fixed orientation. We use three Intel Realsense cameras: two 435 cameras mounted at a side view and on the robot wrist, and one 405 camera mounted overhead. For additional details, see Appendix B.

**Bimanual Tasks.** We consider four bimanual tasks, as shown in Fig. 3, and test the generalization of BUDS to unseen objects (Fig. 2). Each task requires both a high-precision acting policy and a dynamic stabilizing policy that restabilizes multiple times during task execution. We emphasize the complexity of the coordination required of these dexterous tasks. Together, these four tasks represent a wide range of real-world bimanual manipulation tasks, which highlights the prevalence of the stabilizing and acting role assignments. For all tasks, we vary the initial position of all objects over each trial. For more details and videos, see Appendix B and our website.

- **Pepper Grinder.** We grind pepper on three plates in order of color—yellow, pink, then blue as shown in Fig. 3. This task requires restabilizing the pepper grinder over each plate in succession.
- **Jacket Zip.** We zip a jacket by pinning down the jacket's bottom and pulling the zipper to the top. Due to the jacket's deformability, the robot must pin the jacket as close as possible to the zipper. We train all models with a red jacket, and the keypoint models on two more jackets: dark grey and blue. We aim to generalize to light grey and black jackets with different material and zippers.
- **Marker Cap.** We cap three markers in sequence from bottom to top of the workspace. This task requires restabilizing after each marker is capped. We train all policies on red, green, and blue Crayola markers and test generalization with Expo and Redimark markers.
- **Cut Vegetable.** We cut a vegetable half (7-9cm) into four 1-4cm pieces with three cuts. This task requires restabilizing the grasp on the vegetable as each cut is made, as the stabilizing arm should hold the vegetable as close as possible to the cut to prevent tearing and twisting. We train on a jalapeño and test generalization with zucchini halves (15-18cm) and celery sticks (8-10cm).

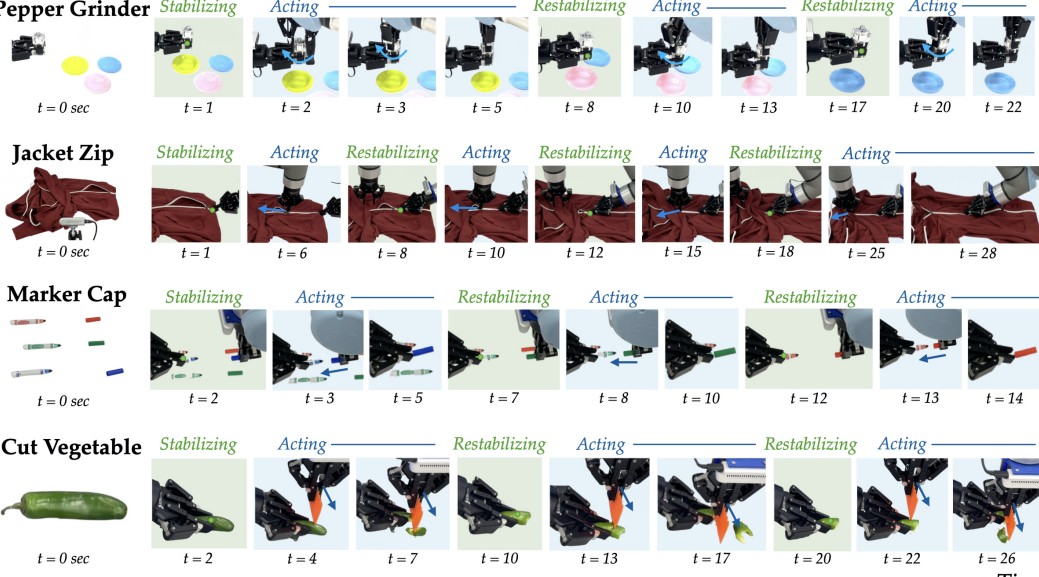

**Pepper Grinder** — *Stabilizing* — *Acting* — *Restabilizing* — *Acting* — *Restabilizing* — *Acting*
t = 0 sec, t = 1, t = 2, t = 3, t = 5, t = 8, t = 10, t = 13, t = 17, t = 20, t = 22

**Jacket Zip** — *Stabilizing* — *Acting* — *Restabilizing* — *Acting* — *Restabilizing* — *Acting* — *Restabilizing* — *Acting*
t = 0 sec, t = 1, t = 6, t = 8, t = 10, t = 12, t = 15, t = 18, t = 25, t = 28

**Marker Cap** — *Stabilizing* — *Acting* — *Restabilizing* — *Acting* — *Restabilizing* — *Acting*
t = 0 sec, t = 2, t = 3, t = 5, t = 7, t = 8, t = 10, t = 12, t = 13, t = 14

**Cut Vegetable** — *Stabilizing* — *Acting* — *Restabilizing* — *Acting* — *Restabilizing* — *Acting*
t = 0 sec, t = 2, t = 4, t = 7, t = 10, t = 13, t = 17, t = 20, t = 22, t = 26

*Time*

Figure 3: **Experiment Rollouts**: We visualize BUDS experiment rollouts. All tasks alternate between updating a stabilizing position $w^s$ while the acting arm is paused and executing an acting policy while the stabilizing arm holds steady. We visualize both $w^s$ and the acting actions.

**Baselines.** **BC-Stabilizer** illustrates the need for a low-dimensional stabilizing representation by replacing the stabilizing keypoint model $f_\theta^k$ with a policy learned from trajectory demonstrations. This policy is instantiated with the same BC-RNN architecture and training procedure as BUDS's acting policy. An oracle classifier determines when BC-Stabilizer has reached a valid stabilizing position, where a noncompliant controller then holds the point stationary as in BUDS while the pre-grasped acting policy from BUDS accomplishes the task. When the restabilizing classifier from BUDS $f_\psi^r$ is triggered, the process repeats. **No-Restable** ablates BUDS's restabilizing classifier and only senses a single sta-

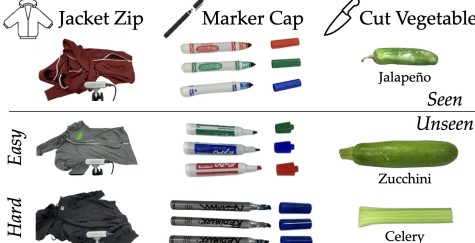

Figure 2: **Task Generalization**: We present the *Seen* and *Unseen* objects in the Jacket Zip, Marker Cap, and Cut Vegetable tasks. We classify these OOD objects into two classes, Easy and Hard, based on their visual similarity to the objects seen during training.

blizing point at the beginning of each task. We evaluate **No-Restable** only on Jacket Zip and Cut Vegetable because other tasks require an updated stabilizing position to reach complete success. We do not compare to a Monolithic baseline (as in Section 3.1) as it achieves zero success for all tasks.

| Task | BC-Stabilizer | No-Restable | BUDS | BUDS Failures | | | |
|---|---|---|---|---|---|---|---|
| | | | | $\hat{w}^s$ | $\pi^a$ | $f_\psi^r$ | $G$ |
| ⚗ Pepper Grinder | $39.9 \pm 21$ | – | $\mathbf{100 \pm 0}$ | 0 | 0 | 0 | 0 |
| 👕 Jacket Zip (Clean) | $28.2 \pm 24$ | $58.8 \pm 39$ | $\mathbf{72.1 \pm 18}$ | 0 | 3 | 3 | 1 |
| 👕 Jacket Zip (Occluded) | $21.6 \pm 17$ | $51.1 \pm 37$ | $\mathbf{55.7 \pm 37}$ | 1 | 2 | 1 | 2 |
| ✏ Marker Cap | $0.0 \pm 0$ | – | $\mathbf{90.1 \pm 16}$ | 1 | 2 | 0 | 0 |
| 🔪 Cut Vegetable | $15.0 \pm 17$ | $46.6 \pm 28$ | $\mathbf{66.8 \pm 24}$ | 2 | 4 | 0 | 3 |

Table 1: **Physical Results:** We report average percent success and standard deviation across 10 trials of 4 bimanual tasks with randomly initialized object positions. For Jacket Zip, we classify initial configurations as Clean or Occluded, where none or up to 30% of the zipper is occluded respectively. We report 4 failure modes: $\hat{w}^s$ stabilizing keypoint, $\pi^a$ acting policy, $f_\psi^r$ restabilizing, and *(G)* poor grasps. We compare to two baselines: BC-Stabilizer where a single-arm IL policy replaces the stabilizing keypoint model, and No-Restable, an ablation of BUDS that disregards restabilizing.

We evaluate BUDS on four bimanual tasks that require dynamic restabilizing. Task success is measured as the proportion of task completed over total amount to be completed, for example zipped length over total zipper length. As shown in Table 1, BUDS achieves 76.9% success across four

tasks, visualized in Fig. 3. We report four failure modes: (1) an incorrect predicted stabilizing position $w^s$, (2) an acting policy failure $\pi^a$, (3) a restabilizing error $f_\psi^r$ that does not detect when a stabilizing point needs updating, and (4) a failed grasp. The acting policy failure is the most common, due to the low amount of data used to train the acting policy and the high precision required. The stabilizing failures ($w^s$ and $f_\psi^r$) are mostly due to large visual differences from the training data, including occlusions, and cause the environment to quickly move out of distribution from the stable, simplified states seen in the acting policy training data. Across all tasks, BUDS outperforms the unstructured BC-Stabilizer baseline due to the high precision required of a stabilizing role. Where BUDS and BC-Stabilizer both learn a relevant point from a visual input, BC-Stabilizer must also learn the policy to reach this position. Thus, the BC-Stabilizer policy's primary failure mode is selecting a poor stabilizing position—it struggles to learn a stabilizing policy robust across many task configurations, as indicated by its 20.9% success rate. BUDS also outperforms No-Restable in Jacket Zip (Clean) and Cut Vegetable, highlighting the need for closed-loop restabilizing. BUDS and No-Restable achieve similar success on Jacket Zip (Occluded) because the biggest challenge in this task is the jacket's deformability and occlusions, which restabilizing alone cannot solve.

| Task | BUDS OOD | | 40-Demo | BUDS Failures | | | |
|---|---|---|---|---|---|---|---|
| | *Easy* | *Hard* | *Hard* | $w_s$ | $\pi^a$ | $f_\psi^r$ | $G$ |
| 👕 Jacket Zip | $62.3 \pm 40$ | $28.8 \pm 27$ | $23.1 \pm 25$ | 10 | 3 | 0 | 2 |
| ✏ Marker Cap | $60.0 \pm 14$ | $53.3 \pm 39$ | $56.7 \pm 39$ | 17 | 1 | 0 | 0 |
| 🔪 Cut Vegetable | $85.0 \pm 13$ | $26.6 \pm 26$ | $30.0 \pm 33$ | 4 | 6 | 0 | 6 |

Table 2: **Generalizability Results:** We test BUDS's robustness to OOD objects of similar morphology. The Easy and Hard OOD objects are respectively more and less similar in visual appearance and dynamics to training objects (Fig. 2). We report average and standard deviation success over ten trials per object, along with failure modes over 20 trials. We compare to 40-Demo, whose acting policy is trained on 40 demonstrations, but do not observe a performance difference on Hard objects. We test BUDS's generalizability to out-of-distribution (OOD) objects classified into two classes based on visual similarity to training objects (Fig. 2). We run 10 trials per object, and find BUDS achieves an average success rate of 52.7% (Table 2). In two of the three tasks, we observe a slight performance drop compared to in distribution settings (Table 1), with a worsening difference for Hard objects. With this expected performance drop, we observe more stabilizing failures ($w^s$ and $f_\psi^r$) due to the stabilizing policy's high visual dependence, which struggles with novel object appearances. For Jacket Zip, we attempt to improve performance by training the stabilizing keypoint model $f_\theta^k$ on three jackets, but the policy still falls short on the vastly different Hard black jacket. 40-Demo aims to improve robustness by training the acting policy on double the data, but again does not significantly improve performance due to the Hard objects' large visual and dynamic differences compared to the training objects, which cannot be remedied with more in-distribution data. We note an exception: Easy zucchini in Cut Vegetable has a higher success rate than that of the in-distribution jalapeño. The hollow jalapeño twists and tears, which is unforgiving of slight acting policy errors, while the solid zucchini can withstand shear forces from noisy policies, yielding more success.

## 6 Conclusion

We present BUDS, a system for dexterous bimanual manipulation that leverages a novel role assignment paradigm: a stabilizing arm holds a point stationary for the acting arm to act in a simplified environment. BUDS uses a learned keypoint as the stabilizing point and learns an acting policy from unimanual trajectory demonstrations. BUDS also learns a restabilization classifier to detect when a stabilizing point should be updated during rollouts. BUDS achieves 76.9% and 52.7% success on four bimanual tasks with objects seen and unseen from training respectively.

**Limitations and Future Work.** Because BUDS uses only visual inputs, it struggles with visually different novel objects unseen during training—BUDS can zip many jackets but struggles with dresses. Thus, BUDS also falls short when tactile feedback is critical, such as plugging in a USB. BUDS assumes fixed roles in each task, which would not hold for tasks where the arms must alternate. In future work, we will explore policies for role assignment, which could be planned to avoid collisions or learned to enable more nuanced tradeoffs. We will incorporate tactile sensing for more sensitive stabilizing, towards tasks like buttoning a shirt.

**Acknowledgments**

This project was sponsored by NSF Awards 2006388, 2125511, and 2132847, the Office of Naval Research (ONR), Air Force Office of Scientific Research YIP award, and the Toyota Research Institute. Jennifer Grannen is further grateful to be supported by an NSF GRFP. Any opinions, findings, conclusions or recommendations expressed in this material are those of the authors and do not necessarily reflect the views of the sponsors. We additionally thank our colleagues who provided helpful feedback and suggestions, especially Suneel Belkhale and Sidd Karamcheti.

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

# Stabilize to Act: Learning to Coordinate for Bimanual Manipulation Supplementary Material

## A  Training Details

We provide details for training each of the models for BUDS: $f_\theta^k$ and $f_\psi^r$ for the stabilizing policy and $\pi_\phi^a$ and $f^g$ for the acting policy.

### A.1  Stabilizing Policy Training

The keypoint models $f_\theta^k$ is trained with a hand-labelled dataset of 30 pairs of images and human-annotated keypoints. We augment each image 10X with a series of label-preserving transformations from ImgAug library [47], including rotation, blurring, hue and saturation changes, affine transformations, and adding Gaussian Noise. The detailed parameters for the transformations are listed in Table 3 and we visualize the image augmentations in Fig. 5. The restabilizing classifier $f_\psi^r$ is trained on a dataset of images from 20 demonstration rollouts with 100 images each. Each image is paired with binary expert annotation of whether

| Augmentation | Parameters |
|---|---|
| LinearContrast | $(0.95, 1.05)$ |
| Add | $(-10, 10)$ |
| GammaContrast | $(0.95, 1.05)$ |
| GaussianBlur | $(0.0, 0.6)$ |
| MultiplySaturation | $(0.95, 1.05)$ |
| AdditiveGaussianNoise | $(0, 3.1875)$ |
| Scale | $(1.0, 1.2)$ |
| Translate Percent | $(-0.08, 0.08)$ |
| Rotate | $(-15°, 15°)$ |
| Shear | $(-8°, 8°)$ |
| Cval | $(0, 20)$ |
| Mode | ['constant', 'edge'] |

Table 3: **Image Data Augmentation Parameters:** We report the parameters for the data augmentation techniques used to train the stabilizing policy's stabilizing position and restabilizing classifier models in BUDS. All augmentations are used from the imgaug Python library [47].

or not restabilizing is needed and augmented by 2X with the same image transformations from above.

Both the keypoint model and the restabilizing classifier are trained against a binary cross-entropy loss with an Adam [50] optimizer. The learning rate is $1.0e^{-4}$ and the weight decay is $1.0e^{-4}$ during the training process. We train these models for 25 epochs on a NVIDIA GeForce GTX 1070 GPU for 1 hour.

### A.2  Acting Policy Training

The acting policy starts from a pre-grasped position, which we achieve using an optional grasping keypoint model. The training procedure of grasping keypoint model $f^g$ is the same as that of stabilizing keypoint model $f_\theta^r$. After the robotic gripper grasps the object, we collect 20 acting demonstration rollouts, each with between 50 and 200 steps. The variation of 20 demonstrations comes from the randomization of initial object position, differences in object shape and dynamics, and variations in grasps. With these demonstrations, we use one set of hyperparameters for all tasks to train a BC-RNN model similar to prior work [51]. We load batches of size 100 with a history length of 20. We learn policies from input images and use a ResNet-18 [46] architecture encoder which is trained end-to-end. These image encodings are of size $64$ and are then concatenated to the proprioceptive input $p_t$ to be passed into the recurrent neural network which uses

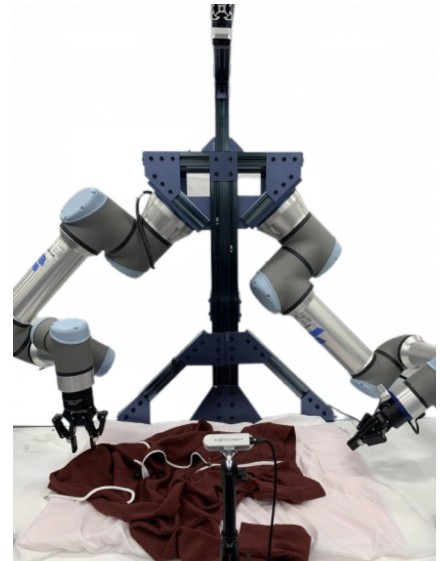

Figure 4: **Experimental Setup**: We present our experimental setup, which uses three cameras due to heavy occlusion during manipulation. One camera is mounted overhead, one is on the wrist of the right arm, and one is facing the front of the workspace at an angle.

a hidden size of 1000. We train against the standard imitation learning loss with a learning rate of

$1e^{-4}$ and a weight decay of $0$. We train for 150k epochs on NVIDIA GeForce GTX 1070 GPU for 16 hrs.

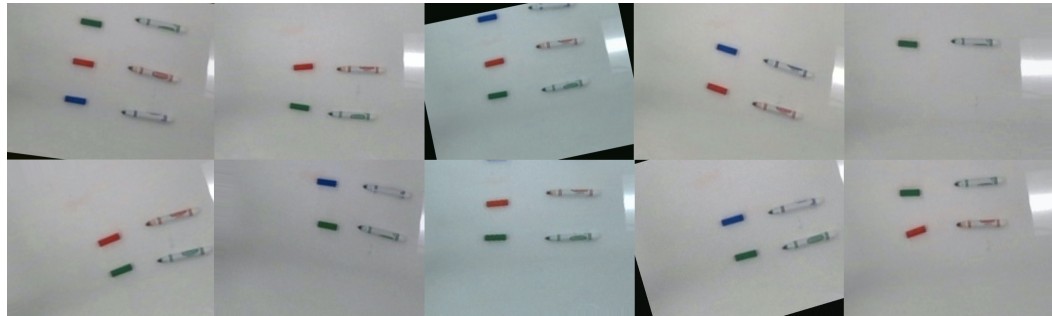

Figure 5: **Data Augmentation for Image Datasets**: We visualize images from the augmentated dataset used to train the stabilizing position model and restabilizating classifier for the marker capping task's stabilizing policy: $f_\theta^k$ and $f_\psi^r$. For $f_\theta^k$, the dataset of expert-labelled image and keypoint annotations is augmented 10X to construct a final dataset of size $300$. For $f_\psi^r$, the dataset is augmented 2X for a final size of $4000$ image and binary classification pairs.

## B Experiment Details

For all tasks, BUDS's acting policy uses a 3D action space. For the three tasks other than Pepper Grinder, this action space represents change in end effector position, $(\Delta x, \Delta y, \Delta z)$. For the Pepper Grinder task, this action space instead represents the change in end effector roll, pitch, and yaw, due to safety concerns involving the closed chain constraint created by using both arms to grasp the pepper grinder tool.

All tasks use the overhead camera for the stabilizing keypoint model and grasping model inputs. Depending on the task and the types of occlusion present during manipulation, we use two of the three cameras for the acting policy and the restabilizing classifier as outlined in Table 4.

We use the optional grasping model $f^g$ for all tasks except the Pepper Grinder task to account for variations of the intial positions of the jacket, markers, and vegetables. Instead for the Pepper Grinder task, the acting arm instead moves to the point corresponding to the end effector position of the stabilizing arm, and grasps at a fixed height above the stabilizing arm corresponding to the height of the pepper grinder. The pepper grinder begins pregrasped in the stabilizing robot hand, but the plate positions are randomly initialized.

| Task | Cameras |
|------|---------|
| Pepper Grinder | Overhead, Side |
| Jacket Zip | Overhead, Side |
| Marker Cap | Overhead, Wrist |
| Cut Vegetable | Wrist, Side |

Table 4: **Task-Specific Cameras:** We report the cameras used for obtaining images as input for the acting policy and restabilizing classifier by task.

In the **BC-Stabilizer** baseline, the stabilizing policy learned via imitation learning is trained with the same procedure as the acting policy for BUDS, with the exception of using an output of two Gaussian mixtures to cover the 3D $(\Delta x, \Delta y, \Delta z)$ action space.

