# OpenReview forum: "Stabilize to Act: Learning to Coordinate for Bimanual Manipulation"
_robot-learning.org/CoRL/2023/Conference — CoRL 2023 Oral_

### Official Review · Reviewer_bbdq · 2023-07-06

**Confidence:** 3
**Originality:** Very Good
**Technical Quality:** Excellent
**Clarity Of Presentation:** Excellent
**Impact:** 4

**Recommendation:**

Strong Accept: I recommend accepting the paper and will argue for my recommendation even if other reviewers hold a different opinion.

**Review:**

The paper is well written, easy to understand, and the motivation and narrative of the paper is quite well done and convincing.
The idea of separating the stabilizing policy from the action policy proves to be beneficial without feeling ad-hoc or limited in scope.
Video was very well done and provides a very nice entry point to the paper.
Perhaps the main merit of the method is the ingenuity of framing a bi-manual task as supporting and acting tasks while not being restricitve on the various types of tasks that a bi-manual robot is expected to solve.
The validation was extensively done, although the task success rate in some cases is a bit disappointing.


**Quality Of The Limitations Section:**

Limitations are addressed clearly

**Questions For Rebuttal:**


I could have missed this in the paper, but what is the frequency of the control loop?

What happens if the stabilizing arm cannot, in fact stabilize? For example, if the vegetable constantly slips from the stabilizing gripper as the acting gripper adds pressure on the object?

The cases of success rates less than 30% is quite disappointing. On the other hand, it seems the number of demonstrations was quite limited. I wonder why the authors decided to stop at only 20 demonstrations, and how the success rate would have increased if more demonstrations were provided.

Not a question, but a comment. It looks like the stabilizing policy could have been profited from methods that explicitly aim at finding invariances among different morphologies such as [1].
[1] P. Florence, L. Manuelli, and R. Tedrake. Self-supervised correspondence in visuomotor policy learning. IEEE RAL, 2019.

Not a question, but another comment. The paper stresses the difficulty of bimanual demonstrations, but there are many teleoperation systems that allow for such tasks. Surely, using a 3D mouse as in the paper is hard, but there are many other examples of bi-manual teleoperation in which complex tasks have been achieved in different ways; from VR headsets/controllers to master-slave mechanisms.



**Robotics Focus:**

Sufficient demonstration on hardware

**Summary Of Paper:**

The paper proposes simplifying the problem of bi-manual manipulation by leveraging the observation that for many tasks of interest, one of the arms is usually used to stabilize an object, while the other is used to act to accomplish the task. Under this premise, the method shows that it is possible to learn a sabilizing policy and then an acting policy sequentially, rather than simultaneously, which greatly simplifes the particular problem of demonstrations.
The method leverages both image-based learning with deep NNs and demonstrations via teleoperation and is evaluated in a series of tasks, using similar objects and out-of-distribution objects.


**Summary Of Recommendation:**

This paper provides some interesting observations on the roles of the arms during bi-manual manipulation and leverages this observation to provide a control solution. The idea is clear, and experiments have been well executed. I think the community will profit from the findings and propositions of the paper.

---

> ### Author Response · Authors · 2023-08-14
> **Response to Reviewer bbdq**
>
> We hope the new experiments doubling BUDS's data budget (40-Demo) in our attached updated paper have addressed your concerns. Because the rebuttal period ends tomorrow, we are following up to see if you have any additional feedback that we can discuss to improve our paper.

---

### Official Review · Reviewer_u9KV · 2023-07-17

**Confidence:** 3
**Originality:** Very Good
**Technical Quality:** Very Good
**Clarity Of Presentation:** Very Good
**Impact:** 4

**Recommendation:**

Weak Accept: I recommend accepting the paper, but will not argue for my recommendation if the majority of other reviewers have a different opinion.

**Review:**

The paper in my opinion is well written and straightforward to understand. The line of argument behind the approach is reasonable and the design decisions made by the authors are well explained and reasonable as well. The approach is evaluated in various experiments and the results seem to be convincing. Citations seem to be adequate, however I am not an expert in this field and therefore can't really tell if important related work has not been cited.

One thing that was not explicitly discussed is if the two robot arms can move independently of each other. The stabilizing policy is not conditioned on the acting policy and at some point will decide to move to a new stabilizing point. From the videos, it looks like the acting policy is stopped during such a movement. The authors also mention the acting policy acts on a stationary environment (l. 224). I'd appreciate if the authors could explicitly clarify if the acting policy is stopped when the stabilizing policy moves to a new keypoint pose.

**Quality Of The Limitations Section:**

Additional details required

**Questions For Rebuttal:**

- If I understood correctly the roles of the arms are fixed. I'd appreciate if the authors could discuss what would be necessary in order to support tasks that require switching the stabilizer/actor roles during a task. I think it is a weakness of the approach that should be addressed in the limitations section.

- Please clarify if the acting policy is stopped while the stabilizing policy is actively moving the robot arm to a new keypoint pose

**Robotics Focus:**

Sufficient demonstration on hardware

**Summary Of Paper:**

The authors present an approach for learning bimanual object interaction tasks from demonstrations. The main idea is to reduce the complexity of the problem by assigning roles to the two robot arms instead of learning a generic bi-manual policy. One arm is assigned the role of a "stabilizer" and has the task of keeping the object to be manipulated in a fixed position. The second arm is assigned the "active" role and has the task of actively manipulating the object. The advantage of the role assignment is that it simplifies the data collection process as well as the learning part due to the reduced overall complexity of the problem.

**Summary Of Recommendation:**

I chose my rating mainly because I did not find any major flaws in the paper.

---

> ### Author Response · Authors · 2023-08-14
> **Response to Reviewer u9KV**
>
> We hope the new implementation clarifications in Section 5 and the additional discussion about role assignment in the Limitations and Future Work section (Sec. 6) in our attached updated paper have addressed your concerns. Because the rebuttal period ends tomorrow, we are following up to see if you have any additional feedback that we can discuss to improve our paper.

---

### Official Review · Reviewer_13Nc · 2023-07-18

**Confidence:** 4
**Originality:** Good
**Technical Quality:** Good
**Clarity Of Presentation:** Very Good
**Impact:** 3

**Recommendation:**

Strong Accept: I recommend accepting the paper and will argue for my recommendation even if other reviewers hold a different opinion.

**Review:**

This paper proposes role assignment to address the high-dimensionality of bimanual tasks, drawing from human inspiration. To help hammer in the difficulty of the problem, the paper explicitly discusses the challenges in trying to solve for the joint two-arm policy. By splitting up the problem into acting and stabilizing, the resulting learning problems are significantly easier. The stabilizing policy is further simplified by considering holding one keypoint "still" and discretely updating this keypoint. This framing is shown to be useful in an experimental comparison that considers learning stabilizing with imitation learning. The acting policy is further simplified by assuming the tasks involves grasping an object and then acting.

The paper proposes four really interesting bimanual tasks: grinding pepper on a series of plates, zipping up a jacket, capping a series of markers, cutting a vegetable in half. These tasks could serve as inspiration to future bimanual manipulation research. Additionally, the method is evaluated in entirely real world experiments which constitutes a significant amount of work (that this reviewer heavily commends!). The video provides a comprehensive summary of the paper and in general the paper is well-written.

Within the paper, the term "stable" is used colloquially as the method does not reason over the forces experienced during the task. This is even though we stabilize, or fixture, an object to hold it still in response to the forces that would perturb the object. For example, consider that for zucchini with a sufficiently large mass, fixturing would not be necessary. The paper would benefit by discussing this limitation and connecting the contribution to previous literature on fixturing (some references given below).

The idea of decomposing the task into acting and stabilizing is a powerful and useful idea that both allows the problem to be tractable and is general enough to work in a variety of settings. Since the stabilizing and acting policy must ultimately work together, it's slightly unclear how necessary information is shared. For example, both the stabilizing keypoint predictor and restabilizing classifier are trained on demonstration images. Given that how an object must be stabilized is conditioned on the details of the task (i.e. where the object is going to be cut), how is the task encoded in the image? The video describes a scenario where the stabilizing point would be updated as the acting arm cuts further up the zucchini but it is not clear how the stabilizing classifier would have access to this information.

Additionally, in choosing where to hold the object still, the stabilizing arm must not only choose a point that stabilizes the object but also a point (and an arm configuration) that enables the acting arm to complete its motion without collision. How is this handled? The keypoint predictor selects a position for the end effector, how is the end effector's orientation and joint configuration selected? In grasping to stabilize an object, how is the grasp force selected? Returning to the need to reason over force requirements, one can imagine situations where a stronger grasp is required to hold an object still.

For the four tasks, task success is measured as the proportion of the task completed. The definition of this is given for the zipper, is it fair to assume that for capping and grinding the success is measured as how many of the three items the robot successfully acts on? How is task success measured for cutting? It seems difficult to precisely measure, in the real world, what percentage of an object has been sliced through.

With respect to related work, [1] considered the idea of decomposing into an acting robot and fixturing robot, specifically in cutting. Both [2, 3] also considered the setting of one agent (a human and robot respectively) acting while the second agent (a robot) fixtures the object, while reasoning how the object needed to be fixtured to resist the exerted forces.
With respect to fixturing generally, both [4, 5] are foundational papers. Finally, [6] also demonstrates how stabilizing an object can make it easier to find an acting policy that operates on that object.

[1] Watanabe, Yoshiaki, Kotaro Nagahama, Kimitoshi Yamazaki, Kei Okada, and Masayuki Inaba. "Cooking behavior with handling general cooking tools based on a system integration for a life-sized humanoid robot." Paladyn, Journal of Behavioral Robotics 4, no. 2 (2013): 63-72.
[2] Chen, Lipeng, Luis FC Figueredo, and Mehmet R. Dogar. "Manipulation planning under changing external forces." Autonomous Robots 44 (2020): 1249-1269.
[3] Holladay, Rachel, Tomás Lozano-Pérez, and Alberto Rodriguez. "Robust Planning for Multi-stage Forceful Manipulation." arXiv preprint arXiv:2208.00319 (2022).
[4] Asada, Haruhiko. "Kinematics of workpart fixturing." In ICRA, vol. 2, pp. 337-345. IEEE, 1985.
[5] Lee, Soo Hong, and M. R. Cutkosky. "Fixture planning with friction." (1991): 320-327.
[6] Shao, Lin, Toki Migimatsu, and Jeannette Bohg. "Learning to scaffold the development of robotic manipulation skills." In ICRA, pp. 5671-5677. IEEE, 2020.

**Quality Of The Limitations Section:**

Additional details required

**Questions For Rebuttal:**

As further detailed above, there are a few questions about how the stabilizing policy operates in tandem with the acting policy: how is the orientation of the end effector chosen? How are the joint angles for the stabilizing keypoint selected such that it is collision-free with respect to the acting policy? How is the stabilizing policy informed about what it needs to stabilize with respect to? This is with respect to how the acting policy is interacting with the object, considering both geometry and force.

**Robotics Focus:**

Sufficient demonstration on hardware

**Summary Of Paper:**

To make learning bimanual manipulation tractable, this paper proposes decomposing tasks such that one arm stabilizes an object while the other acts on it. Rather than having to learn a monolithic joint policy, and to collect complex data, this strategy enables the learning of a separate stabilizing policy and acting policy. Stabilizing is defined as statically holding an object in place, readjusting when needed. Thus the stabilizing policy is composed of a keypoint predictor, for where to hold, and a classifier on when to readjust. The acting policy can then be learned in the simpler environment where the object is assumed to be stabilized. Since acting is defined as grasping an object and then moving, the policy is composed of a grasping keypoint predictor and imitation learning. The paper considers 4 bimanual tasks and evaluates the results in many real world demos. Ablation studies show that the method struggles with objects visually different from the training data.


**Summary Of Recommendation:**

This paper focuses on the very complex and interesting task of bimanual manipulation and proposes separating the problem into stabilizing and acting. The paper demonstrates the method on four well-chosen tasks and conducts a large quantity of real world experiments. The paper would strongly benefit from clarifying how the stabilization method reasons over force (and if it doesn't, why this is sufficient) and how it generates solutions that are compatible with the acting policy.

[My recommendation score has been updated as a result of the rebuttal. I have also added a comment below]

---

> ### Author Response · Authors · 2023-08-14
> **Response to Reviewer 13Nc**
>
> We hope the new "Stabilizing in Manipulation" Related Work section, the implementation clarifications, and the updated force discussion in the Limitations and Future Work section in our attached updated paper have addressed your concerns. Because the rebuttal period ends tomorrow, we are following up to see if you have any additional feedback that we can discuss to improve our paper and, given our aforementioned paper updates, we ask that you consider raising your score.

---

### Official Review · Reviewer_ku5C · 2023-07-21

**Confidence:** 4
**Originality:** Good
**Technical Quality:** Good
**Clarity Of Presentation:** Good
**Impact:** 3

**Recommendation:**

Weak Accept: I recommend accepting the paper, but will not argue for my recommendation if the majority of other reviewers have a different opinion.

**Review:**

Pros
The paper is well written and accompanied by a video that does a good job at explaining the general approach. The real-robot demo videos look cool and the diversity of tasks is a strong positive.

Cons
The proposed method is only compared to one baseline. Given the complexity of the system, I think the authors should compare to ablated versions of their system. Additionally, the one baseline that is evaluated is not exactly a fair comparison. BUDS is provided with a lot of additional task and problem specific human annotations (such as keypoints, pregrasp points) that the BC baseline isn't given.

Although the method improves data efficiency for certain tasks, it also greatly increases the amount of expert annotations needed, resulting in fewer total datapoints per human expert hour.

**Quality Of The Limitations Section:**

Limitations are addressed clearly

**Questions For Rebuttal:**

How does the stabilization controller work? The authors only explain how the keypoint detection model is trained, but how is the robot actually controlled to stabilize a given keypoint? Does the hand move to reposition the keypoint if the object slips in the hand?

What are the primary reasons for failure for BC-Stabilizer? A video was only shown for the cutting task and it looked like the gripper fingers bent and the pepper slipped. Could something like this be fixed with a stiffer controller? Are there other failure modes for the BC baseline?

**Robotics Focus:**

Sufficient demonstration on hardware

**Summary Of Paper:**

This paper introduces BimanUal Dexterity from Stabilization (BUDS), a strategy for bimanual robotic manipulation. The authors propose a division of labor between one stabilizing hand that keeps an object fixed in place and one active hand that interacts with the stabilized object in some way. The approach relies on several keypoint-based perception networks that determine where to grasp objects, what point on the object to hold stable, and when to switch to stabilizing a new object keypoint. The results show an improvement over behavior cloning with no explicit stabilization constraint.

**Summary Of Recommendation:**

Overall, the paper is well-presented and easy to understand with good robot demo videos. Additionally, it shows large improvement over a commonly-used behavior cloning baseline on the selected tasks. However, I think the paper could use some additional ablations and a few fairer baseline comparisons that make use of the extensive human annotations. The paper is also lacking explanation on some aspects of the system and a description of the failure modes of behavior cloning.

---

> ### Author Response · Authors · 2023-08-14
> **Response to Reviewer ku5C**
>
> We hope the new BUDS ablation experiments (No-Restable and 40-Demo) in addition to the implementation clarifications in our attached updated paper have addressed your concerns. Because the rebuttal period ends tomorrow, we are following up to see if you have any additional feedback that we can discuss to improve our paper.

---

### Author Response · Authors · 2023-08-12
**Summary Response to Reviewers**

We thank everyone for their thoughtful reviews, and are excited that the reviewers agree that a stabilizing and acting role assignment is a clever approach to adding structure for a wide variety of complex bimanual tasks. In addition to the individual responses for each reviewer, we summarize and discuss common concerns here.

_Additional Method Comparisons_ (Rev ku5C, Rev bbdq) - Two reviewers requested additional ablation experiments for BUDS. We provide two ablations: No-Restable (Table 1), which is BUDS without a restabilizing component, and 40-Demo (Table 2), which doubles the number of demonstrations used to train BUDS’s acting policy. We observe from the No-Restable comparison that the restabilizing component is indeed necessary for precise execution required of the Jacket Zip and Cut Vegetable tasks. While the 40-Demo method was motivated in an effort to boost BUDS’s success rates in OOD object settings, we do not observe a difference in performance compared to BUDS. This is because the difficulty in the OOD settings are the large visual and dynamic differences between the in-distribution and OOD objects (i.e. red vs. black jackets or cracking jalapenos vs. fibrous celery), which additional in-distribution demonstrations cannot address.

_Additional System Details_ (Rev ku5C, Rev 13Nc, Rev u9KV) -  Multiple reviewers had questions about the robot controllers and collision avoidance. We apologize for the confusions about this part of the system. Both robot arms are controlled at a frequency of 10 Hz and grasp with a force of 225N. All robots move along a linear path between all points, and calculate joint poses using the IK solver from [1]. While the stabilizing arm is moving between restabilizing points, the acting arm is held stationary in the home position to avoid the need for planning for collision avoidance. We have included all these details to the paper main text in Sections 4 and 5.

_Force Sensing for Stabilizing_ (Rev 13Nc) - One reviewer suggested the need for force sensing for a robust stabilizing policy, and we agree that force is a useful form of feedback in a stabilizing setting. In this work, we found that using only a vision-based stabilizing policy is able to achieve high success across a wide variety of bimanual tasks, and thus are able to sidestep the additional complexity and hardware that incorporating forces entails without sacrificing task complexity. We agree that incorporating forces into the stabilizing policy is an interesting direction for future work for tasks with more occlusion or precision.

We present a list of new changes and updates to the paper below:

- Ablations of BUDS without a restabilizing component (Table 1)

- Ablations of BUDS number of demonstrations (Table 2)

- New “Stabilizing in Manipulation” section within Related Work (Sec. 2)

- Additional robot controller details (Sec. 5)

- Additional Limitations and Future Work (Sec. 6)

We have incorporated all additional experiments and reviewer suggestions into the updated draft in blue. Thank you!

[1] https://sdurobotics.gitlab.io/ur_rtde/

---

### Decision · Program_Chairs · 2023-08-30

**Decision:**

Accept (Oral)

**Comment:**

I agree with the reviewers highly positive remarks on this paper. The authors also did a great job in their rebuttal responding to the minor concerns brought up by the authors. I recommend the paper be accepted.